# Explicit description of viral capsid subunit shapes by unfolding dihedrons
Ryuya Toyooka[1], Seri Nishimoto [1], Tomoya Tendo[1], Takashi Horiyama[2] ✉, Tomohiro Tachi [1] ✉ & Yasuhiro Matsunaga [3] ✉

Viral capsid assembly and the design of capsid-based nanocontainers critically depend on understanding the shapes and interfaces of constituent protein subunits. However, a comprehensive framework for characterizing these features is still lacking. Here, we introduce a novel approach based on spherical tiling theory that explicitly describes the 2D shapes and interfaces of subunits in icosahedral capsids. Our method unfolds spherical dihedrons defined by icosahedral symmetry axes, enabling systematic characterization of all possible subunit geometries. Applying this framework to real $T = 1$ capsid structures reveals distinct interface groups within this single classification, with variations in interaction patterns around 3-fold and 5-fold symmetry axes. We validate our classification through molecular docking simulations, demonstrating its consistency with physical subunit interactions. This analysis suggests different assembly pathways for capsid nucleation. Our general framework is applicable to other triangular numbers, paving the way for broader studies in structural virology and nanomaterial design.

Viral capsids are assemblies of proteins that encapsulate and protect the viral genome. Many spherical viral capsids adopt icosahedral structures, which is fully characterized by 60 symmetry operations. To elucidate the mechanism of self-assembly of molecules[1,2] for the rational design of capsid-based nanocontainers[3–6], it is important to understand how icosahedral symmetry imposes geometric constraints on the interaction patterns between subunit proteins.

The Caspar-Klug (CK) theory is currently used as a major tool for classifying capsid structures[7]. The theory explains how capsids can be formed from different numbers of subunits, resulting in various sizes[8]. In the CK theory, protein subunits are modeled using a hexagonal network of subunits on a plane according to the *p6* wallpaper group, whose 6-fold symmetric interface on the plane is regarded as quasi-equivalent to that of the 5-fold symmetry axis on the icosahedron. The CK theory systematically describes the size and number of subunits by subdividing the triangular region into multiple triangles using two integers $(h, k)$ and the triangulation $(T)$ numbers. On the other hand, the CK theory cannot directly address questions on what the necessary shapes of subunits and interfaces for self-assembling icosahedral capsids are. This question is not only essential for the rational design of capsids of subunit proteins but also for understanding the possible formation pathways during the self-assembling process. To answer this question, an explicit description or characterization of all possible subunit shapes or interfaces would be needed.

Thus far, several contributions have directly or indirectly focused on characterizing the shapes of subunits in viral capsids from viewpoints beyond the CK theory[9–15]. These studies include affine extensions to describe the icosahedral groups and virus capsids[10,12], and a framework to investigate the locations of protrusions[13]. Twarock introduced non-triangle tiles instead of the CK theory's triangular tiles to characterize Simian Virus 40 and L-A virus capsids[9]. Raguram et al. also developed a general polyhedral framework to describe virus capsid structures, employing pentagonal subunits, accounting for intrinsic capsid chirality[14]. Twarock and Luque recently elegantly extended the CK theory by using non-CK tiles (Archimedean tiles and their duals) to describe capsid structures that fall outside the CK description[15]. While these works significantly broadened the spectrum of describable capsid geometries beyond the original CK theory, the subunit shapes that can be systematically handled are still confined to specific geometry forms, such as typical pentagonal, hexagonal, and triangular tiles, as well as rhombs, kites, and florets in the dual representations.

The contribution of this study is to develop a framework for describing the shapes and interfaces of subunit proteins based on a novel representation using the tiling theory on spherical surfaces. The idea is based on the unfolding of dihedrons with the triangular shape used in the CK theory. Our representation can account for all possible 2D shapes and interfaces of subunit proteins of icosahedral capsids. Using the proposed representation, we classify the icosahedral structures according to the shapes and interfaces

[1]Department of General Systems Studies, The University of Tokyo, Tokyo, Japan. [2]Faculty of Information Science and Technology, Hokkaido University, Sapporo, Japan. [3]Graduate School of Science and Engineering, Saitama University, Saitama, Japan. ✉e-mail: horiyama@ist.hokudai.ac.jp; tachi@idea.c.u-tokyo.ac.jp; ymatsunaga@mail.saitama-u.ac.jp

of their subunits in the $T = 1$ group. Then, we show that there are distinct interface types within the same $T = 1$ group, with variations in interface curve lengths around the 5-fold, 3-fold, and 2-fold symmetry axes. We propose these interface curve lengths as a governing factor to classify the interactions between subunits. To validate our classification and understand the physical interactions between subunits, we employ pairwise docking simulations. These simulations investigate the potential interactions between two protein subunits, predicting their most favorable binding orientations and energies. By comparing the results of these simulations with our geometric classification, we can assess the consistency between the predicted physical interactions and the structural patterns identified by our framework. These results imply that there are multiple types of assembling interactions in capsid structures and a strategy for creating a nucleus in the self-assembly process.

## Results

### Dihedron-unfolding model based on spherical tiling

We focus here on $T = 1$ tiling for simplicity, although our approach is general and can straightforwardly extend to $T \neq 1$ cases (see Supplementary Fig. 1 and text for extension). In $T = 1$ tiling, 60 subunits are assembled according to chiral icosahedral symmetry. Subunit shapes can be generated by creating the fundamental figure or *tile* that makes up 1/60 of the sphere (Fig. 1b). The boundary of the tile can be segmented into at most three pairs of identical curves copied by rotations of 180°, 120°, and 72° about the 2-, 3- and 5-fold symmetry axes, respectively (Fig. 1a, b).

Consider the tile as a thin, flexible sheet on the sphere's surface. The symmetry axes of the icosahedron ensure that when we stitch the identical boundary curves together, they align perfectly (Fig. 1b, c). As we stitch along each symmetry axis, the vertices of the tile are brought closer together. The final stitching necessarily brings all three vertices to meet at a single intersection point. In the end, this stitching transforms the original tile into a *dihedron*[16]: a doubly-covered half equilateral triangle that covers two adjacent triangular regions. This dihedron is a 2-to-1 mapping precisely covering 1/120 of the sphere's surface.

By considering its inverse process, any tile figure can be obtained; i.e., we start from the dihedron, cut it along an arbitrary curve, and unfold it into a spherical tile (Supplementary Movie 1). To unfold the triangular dihedron into a tile, the dihedron must be cut along a Y-shaped path (cut tree) starting from a single point we call the *junction point* with three ends at the vertices (the cut tree and the junction point shown in Fig. 1b, c). To focus on the contact relationship, we here simplify the cut to three shortest paths (geodesics) on the sphere's surface (Fig. 1c). We construct a polygonal tile by alternately connecting the three symmetry axes and the three mirror images of the junction point relative to the edges of the spherical triangle. Under this simplification, the unfolding is uniquely characterized by the choice of orientation of the dihedron and the location of the junction point for the cutting. The orientation corresponds to choosing either a left- or right-

handed triangle as the basis for the dihedron (Here, the triangle with vertices 2, 3, and 5 in counterclockwise order is called the right-handed).

This dihedron-unfolding model provides two-parameter (the location of the junction point and the orientation) representation of how subunits connect with each other in $T = 1$ group. Figure 2 shows tile shapes and connections for various positions of the junction point and the orientation. Note that the junction point can be placed outside the triangle. Even in this case, the shape of the tile can be defined similarly by mirror reflection with respect to each edge, resulting in a concave tile (Fig. 2 and Supplementary Movies 2 and 3). The junction point must be within the highlighted area in Fig. 2; otherwise, the unfolded tile self-intersects (i.e., mirror-reflected edges intersect with each other).

The connectivity between the tiles can be represented by the polyhedral graph (black lines in Fig. 2) and its dual graph (red dashed lines in Fig. 2). In the dual graph representation, each point (node) represents a subunit, and lines (edges) between points show connections between subunits. For visualization purposes, we use mapping from a sphere onto a flat surface (gnomonic projection) that represents shortest paths (geodesic lines) on the sphere's surface as straight lines. This lets us specify the junction point on a projected flat plane, which we set perpendicular to the 2-fold symmetry axis ($z$-axis in Fig. 2).

In a *generic case*, when the junction point is not on any symmetry axes, each tile becomes pentagonal. Each subunit is connected to five neighboring subunits: one through 2-fold symmetry, two through 3-fold symmetry, and two through 5-fold symmetry. The polyhedral graph of this tiling is classified to the pentagonal hexecontahedron (a 60-faced polyhedron with pentagonal faces); its dual graph is the snub dodecahedron (a polyhedron with 12 pentagonal and 80 triangular faces). This case includes the pentagonal tiles investigated by Raguram et al.[14] in their study to develop a general polyhedral framework. As already discussed by Raguram et al.[14], the pentagonal hexecontahedron and its dual have orientation (chirality), which depends on the choice of orientation of the triangle in our dihedron-unfolding model. The left- and right-handed tiles (also called Laevo and Dextro, respectively) result in left- and right-handed pentagonal hexacontahedral graphs, respectively.

In *degenerate cases*, we locate the junction point on the 2-, 3-, and 5-fold symmetry axes. When the junction point is on the 2-fold symmetry axis, we obtain quadrangular tiles classified to the deltoidal hexecontahedron (a 60-faced polyhedron with kite-shaped faces); its dual is classified to the rhombicosidodecahedron (a polyhedron with 20 triangular, 30 square, and 12 pentagonal faces). Each subunit is connected to four neighboring subunits: two through 3-fold symmetry and two through 5-fold symmetry. This case includes the kite-like tiles studied by Twarock and Luque[15] for the analysis of Tobacco ringspot virus[17] using non-CK tiles. When the junction point is on the 3-fold symmetry axis, we obtain triangular tiles classified to the pentakis dodecahedron (a 60-faced polyhedron with triangular faces); its dual is classified to the truncated icosahedron (a polyhedron with 12

**Fig. 1 | Dihedron unfolding and resulting tile shapes. a** A dihedron (represented by an orange triangle), which is a double covering of the spherical triangle formed between the three axes of symmetry (1/120 of the sphere's surface). The cut tree is shown by solid red lines. **b** The unfolding of the dihedron, forming a tile (outlined by solid red lines) using the cut tree (indicated by dashed red lines). **c** Another example where the cut tree corresponds to three geodesics (shortest paths on the sphere's surface).

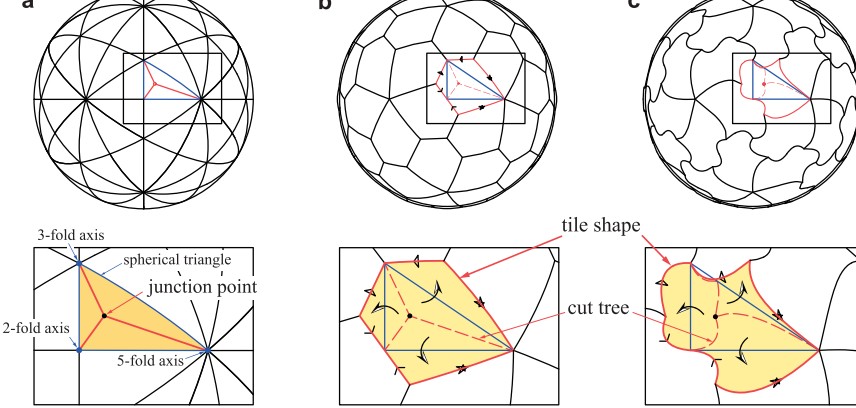

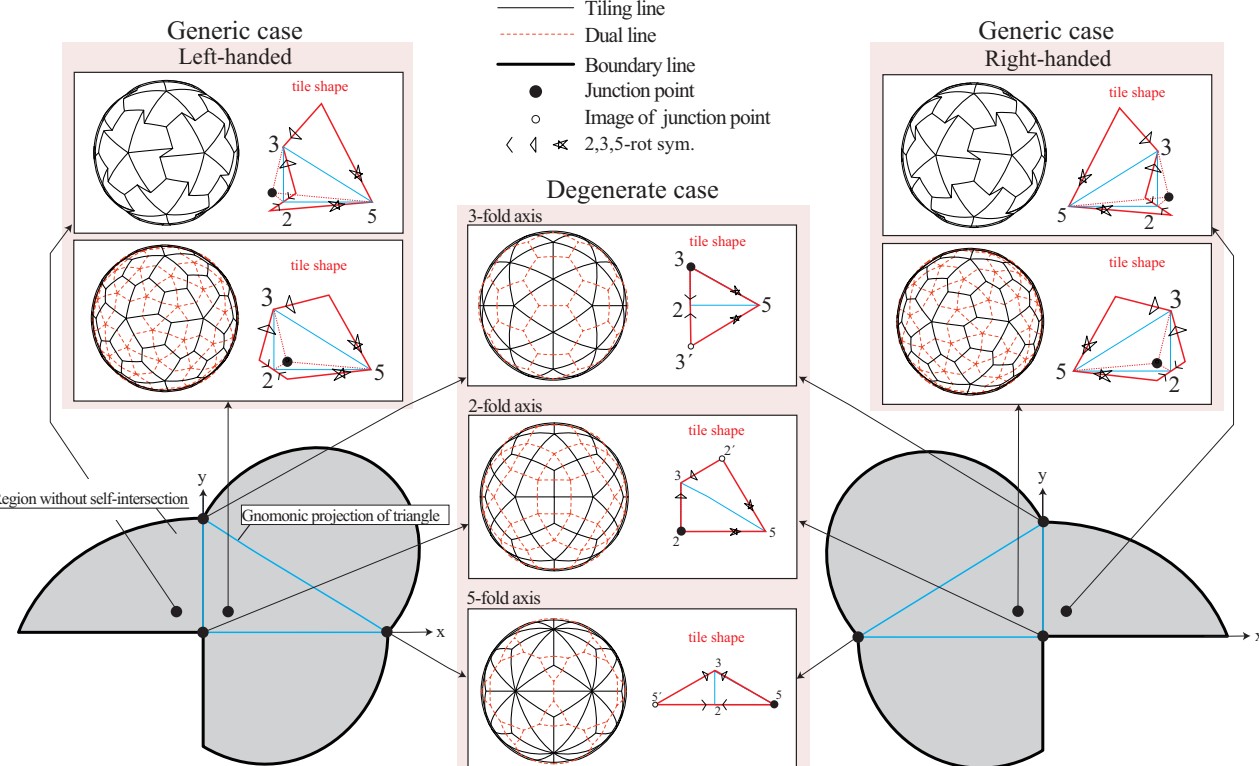

**Fig. 2 | Dihedron-unfolding model explicitly describing various tile shapes and their contacts.** Triangles outlined blue solid lines represent gnomonical projection of spherical triangles (dihedrons). The gray areas represent the allowed positions for the junction points. If the junction point is placed outside this gray area, the resulting unfolded shape intersects itself. In generic cases, the resulting tile has a left- or right-handed orientation (chirality). When the junction point is placed outside the triangle but still within the gray area, the resulting tile becomes concave. In special cases (degenerate cases), when the junction point is exactly on one of the symmetry axes (the vertices of the triangle), the resulting tile lacks chirality (is achiral). Each small diagram shows how different junction point positions lead to different tile shapes and contact patterns (represented by broken red liens) between neighboring tiles in the final capsid structure.

pentagonal and 20 hexagonal faces). Each subunit is surrounded by three neighboring subunits: on through 2-fold symmetry and two through 5-fold symmetry. When the junction point is on the 5-fold symmetry axis, we obtain triangular tiles classified to the triakis icosahedron (another 60-faced polyhedron with 60 triangular faces); its dual is classified to the truncated dodecahedron (a polyhedron with 12 ten-sided and 20 triangular faces). Each subunit has three neighbors: one through 2-fold symmetry and two through 3-fold symmetry.

## Characterization of real capsid structures

We fitted our dihedron-unfolding model to experimental capsid structures of $T = 1$ to classify real structures. Here, by fitting the model to real structures, we inversely estimated the left- or right-handed orientation and location of the junction point for each capsid. We first determined left- or right-handed orientations based on the observation that the boundary of the tile passes through all three axes. So, we chose the orientation such that the maximum of the shortest distances from the three axes to the atom positions are smaller (see Fig. 3a for the interpretation of the criteria). Then, we calculated the fitness (measured by the Dice coefficient[18]) between the real structures (silhouette represented by the circles with the effective radii of amino acids centered at their Cα atom positions of subunit protein) and the tile in the gnomonic projection and maximized the fitness using a genetic algorithm (see Methods for details).

Figure 3 illustrates the computed orientations and locations of the junction points for the real capsid structures. We used the capsid structures of $T = 1$ number taken from Protein Data Bank (PDB) whose PDB IDs are 2BUK[19], 4V4M[20], 6S44[21], 7ODW[22], 3R0R[23], 5ZJU[24], 1STM[25], and 1VB4[26]. PDB IDs 2BUK and 4V4M belong to the satellite tobacco necrosis virus, both sharing identical sequences (structures are slightly different due to experimental conditions). Also, PDB IDs of 3R0R and 5ZJU belong to the porcine circovirus 2, albeit with slightly different sequences. This selection of redundant pairs aimed to check the robustness of our fitting procedure. The other structures have different origins and sequences with each other: the faba bean necrotic stunt virus (PDB ID 6S44), a model of the haliangium ochraceum encapsulin (PDB ID 7ODW), and the sesbania mosaic virus deletion mutant (PDB ID 1VB4). The structure of the L-A virus (PDB ID 1M1C[27]), classified as $T = 2$ comprising a 120-homomer, was also included. In this capsid, a neighboring subunit pair (dimer) was treated as a single tile in the fitting process. For details of the structural data, see Methods and Supplementary Fig. 2.

In Fig. 3b, all capsids fall into generic types, where PDB IDs 2BUK, 6S44, 7ODW, 4V4M are left-handed and 1VB4, 1STM, 5ZJU, 3R0R, and 1M1C are right-handed. In the figure, we provide a simple metric based on the lengths of shared edges (interface) between neighboring capsids copied by the 2-, 3- and 5-fold axes to evaluate the contacts between adjacent subunits numerically. These lengths are the distances from the axis for 3- and 5-fold axes, while twice the distance is used for the 2-fold axis as we consider the edges active in a face-to-face manner between a pair of subunits. As the longer contact edge is associated with the strong interaction between the subunits, the categorization with this measure suggests the strength of pairwise interaction between the subunits, which we verify in the following section. The contour plot shows the maximum value of the distances; this decomposes the region into three domains governed by contact with respect to the 2-, 3-, and 5-fold axes.

The figure shows that the subunits of the satellite tobacco necrosis viruses (PDB IDs 2BUK and 4V4M), which have sequences identical to each other, possess the longest contact edge between their copies along the 5-fold axis in the left-handed. In contrast, the other twin subunits, porcine

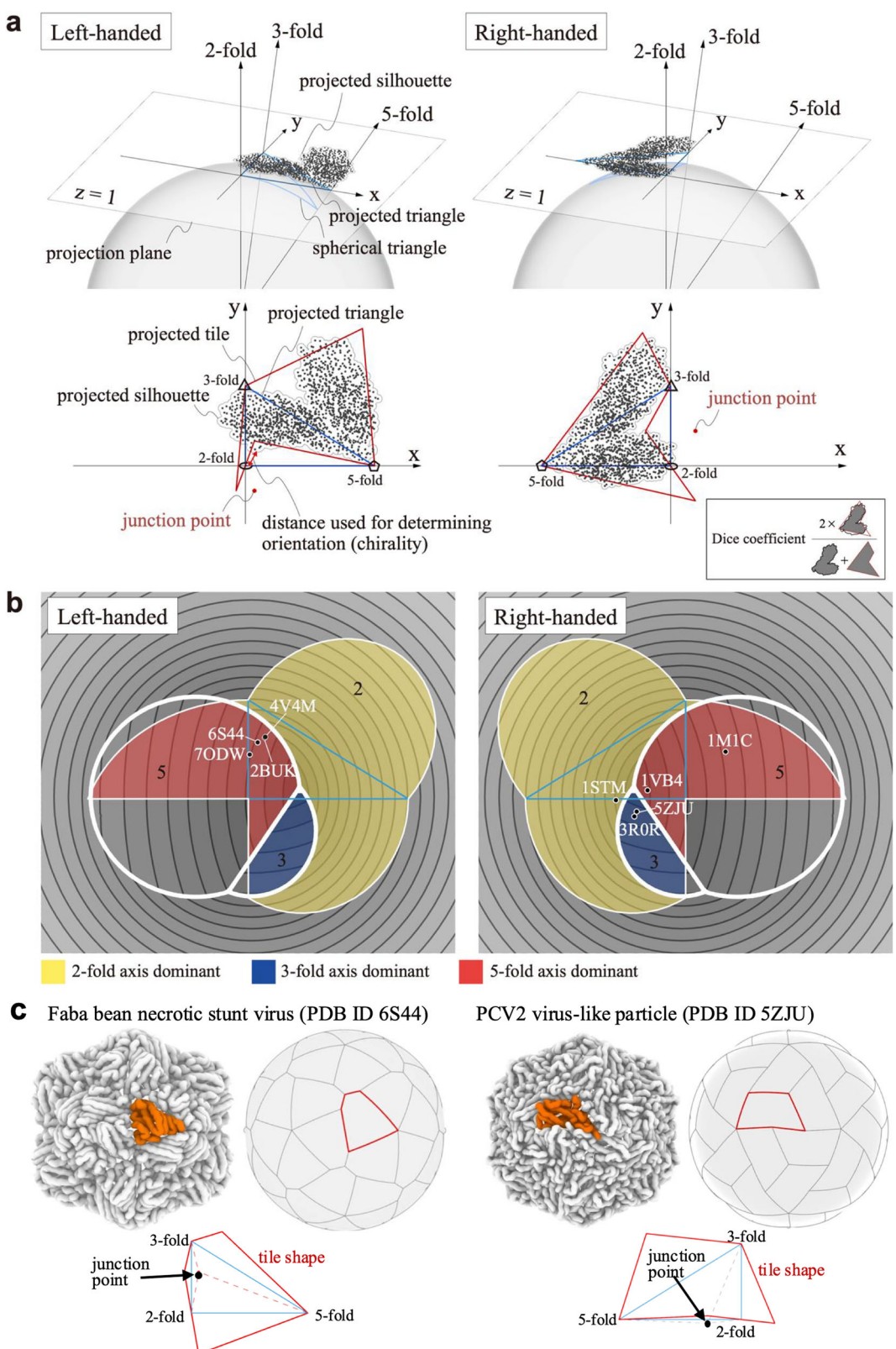

**a**

Left-handed / Right-handed

2-fold · 3-fold · projected silhouette · 5-fold · z = 1 · y · x · projected triangle · spherical triangle · projection plane

projected triangle · projected tile · projected silhouette · 3-fold · junction point · 2-fold · 5-fold · junction point · distance used for determining orientation (chirality)

Dice coefficient

**b**

Left-handed: 4V4M, 6S44, 7ODW, 2BUK — regions 5, 2, 3
Right-handed: 1M1C, 1STM, 1VB4, 5ZJU, 3R0R — regions 2, 5, 3

2-fold axis dominant · 3-fold axis dominant · 5-fold axis dominant

**c**  Faba bean necrotic stunt virus (PDB ID 6S44) · PCV2 virus-like particle (PDB ID 5ZJU)

3-fold · junction point · tile shape · 2-fold · 5-fold

3-fold · junction point · tile shape · 5-fold · 2-fold

circoviruses 2 (PDB IDs 3R0R and 5ZJU), with almost identical sequences to each other, exhibit the longest contact edge around the 3-fold axis, characterized by a different orientation (the right-handed). Interestingly, other subunits also cluster around the same regions as these twin subunits, namely the left-handed 5-fold axis and the right-handed 3-fold axis. Since there is no physical reason to prefer any specific orientation for subunit interactions,

the observed tendency for these junction points to cluster would be evolutionarily coincidental. Another intriguing observation is that almost all junction points do not distribute around regions governed by the 2-fold axis (except for the satellite panicum mosaic virus, PDB ID 1STM). The 2-fold symmetry that interacts in a face-to-face manner is evolutionarily easier to optimize compared to other symmetry axes[28]. In fact, it is well known that

**Fig. 3 | Analysis of real capsid structures using our dihedral unfolding framework. a** Fitting of the junction point location to a real capsid structure. Gnomonic projection of a subunit is shown for the L-A helper virus (PDB ID 1M1C). Black dots represent the Cα atom positions of the subunit protein. Arrows labeled 2-fold, 3-fold, and 5-fold indicate the projected 2-fold, 3-fold, and 5-fold symmetry axes, respectively. The blue line outlines the spherical and projected triangles (dihedrons). Red dots represent the locations of the junction points on the projected plane that best matches our model to the real structure. The orientation (chirality) of tile is determined according to the distances between the symmetry axes and the Cα atom positions. The junction point location is optimized by maximizing the Dice coefficient. **b** Results of fitting our model to various real capsid structures. Left and right panels show left-handed and right-handed orientations. Colored regions indicate where the junction point does not induce self-intersection, with yellow, blue, and red areas corresponding to domains governed by contact with respect to the 2-, 3-, and 5-fold axes, respectively. Capsid structures (black circles) are labeled with their PDB IDs. **c** Schematic representations of subunit-subunit interactions for 3-fold axis dominant (the faba bean necrotic stunt virus, PDB ID 6S44), and 5-fold axis dominant (the PCV2 virus-like particle, PDB ID 5ZJU) cases. Real subunit shapes are colored by orange, and fitted tiling shapes are shown with red lines. The junction points are located by black arrows.

structures stabilized around the twofold axis are common in ordinary dimers[29]. On the other hand, in the case of capsids, symmetries other than the 2-fold may be more important for creating a kinetic nucleus for growth toward full capsid formation.

### Pairwise docking simulation of subunit proteins

In order to investigate whether our classification of subunit contacts is consistent with physical interactions, we conducted rigid-body docking simulations of paired monomeric subunits. In the rigid-body docking simulations, the structure of the subunit protein is treated as a rigid body, and no conformational changes are considered. We here used ZDOCK[30,31] for the simulation. In ZDOCK, physical and statistically derived interaction energies are approximately calculated with regular grids and docking poses (translations and orientations) with high docking scores are exhaustively searched.

Figure 4 shows the results of the docking simulations. Here, to relate the docking poses to the symmetry axes of capsid, we calculated the screw axis[32] for individual docking structure. The screw axis is an axis that describes a rigid body motion (translation and rotation) to superimpose one monomer to the other one (Supplementary Fig. 5). The screw axis and translation and rotation around that axis were determined by using the Rodriguez equations (see Methods). The maximum scores of docking scores and the minimum values of root mean square deviation (RMSD) from the experimental structure are shown as heatmaps in the space of rotations and translations of the screw axes.

The figure shows that the sesbania mosaic virus capsid (PDB ID 1VB4, classifiled as right-handed and 5-fold in our dihedron-unfolding model) and the faba bean necrotic stunt virus (PDB ID 6S44, left-handed 5-fold) have high docking scores at around the axes of the 5-fold (72° in the figure) and 2-fold axis (180°). Among these two axes with high docking scores, the 5-fold axis was confirmed to be indeed stable based on RMSD, implying the consistency with the classification with the spherical tiling. This consistency was also confirmed for the other left-handed 5-fold capsids, the satellite tobacco necrosis viruses (PDB IDs 2BUK and 4V4M, Supplementary Figs. 6 and 7). The porcine circovirus 2 like particle (PDB ID 5ZJU, right-handed 3-fold) does not have stable docking structures at around the 5-fold axes. Instead, it has stable structures around the 3-fold axis (120° in the figure) and the 2-fold axis. The 3-fold axis was confirmed to be indeed stable from the result of RMSD. Additionally, the satellite panicum mosaic virus (PDB ID 1STM, classified as right-handed 2-fold) exhibits a single docking score peak at around the 2-fold axis. The structure around the 2-fold axis is consistently validated by the RMSD result.

The model of haliangium ochraceum encapsulin (PDB ID 7ODW), which is classified as left-handed and 5-fold, shows an exceptional result (Supplementary Figs. 6 and 7). In this case, we could not find notable docking score peaks except for the 2-fold axis. The reason would be that this capsid has interactions where subunits stack on top of each other in a direction perpendicular to the spherical surface (Supplementary Fig. 2), which would not be adequately characterized by 2D tiling. Also, the L-A virus (PDB ID 1M1C), which treats two subunits as one tile, shows exceptional behavior. While our model classifies it as right-handed 5-fold, the docking simulation results indicate that the 3-fold and 5-fold axes are stabilized to a similar degree (Supplementary Figs. 6 and 7). This suggests

that our model may be underestimating the contribution of interactions due to the concave shape formed around the 3-fold axis, which is exceptionally created by combining two subunits.

As for references, we further performed the same type of docking simulation for non-capsid structurtes: the tobacco mosaic virus subunit (PDB ID 6R7M, Fig. 4d) that only has a 16 1/3 symmetry axis and an NMR structure of chymotrypsin inhibitor structure in solution (PDB ID 2M99, Supplementary Figs. 6 and 7) that does not have any symmetry axes. Our docking simulation correctly captured the stable symmetry axis for the tobacco mosaic virus and do not show notable peaks except for 2-fold axis in the case of chymotrypsin inhibitor.

The initial stages of capsid self-assembly involve complex interactions between subunits. Our pairwise docking simulation results indicate that there may be preferences in subunit interactions near different symmetry axes depending on the subunit shape. Notably, our results suggest that dimers around the 5-fold or 3-fold axes can be stabilized without the presence of other neighboring contacts. This finding provides insights into the potential early stages of capsid assembly, although it does not necessarily imply that dimer formation is the most important step in the process.

It is noted that capsid assembly is influenced by multiple forces beyond simple hydrophobic interactions. As recently studied by Panahandeh et al.[33], elastic energy from protein stretching and bending significantly affects the assembly process and final capsid structure. These elastic forces, along with hydrophobic interactions and other factors such as protein concentration, contribute to the energy barriers in capsid formation and help guide the assembly towards symmetrical structures. For capsids with T numbers greater than 1, the assembly process generally involves the formation of both pentamers and hexamers. While pentamers are necessary due to the spherical geometry of the capsid, hexamers are generally preferred by hydrophobic interactions. The interplay between these structures is crucial for the final capsid formation.

### Discussion

In this study, we have proposed a novel framework based on spherical tiling theory to explicitly describe the possible 2D shapes and interfaces of subunits in icosahedral capsids. This approach has allowed us to classify $T = 1$ real capsid structures in terms of subunit shapes and interfaces. Our findings reveal distinct interface groups even within the same $T = 1$ classification, highlighting variability in interaction patterns around 3-fold and 5-fold symmetry axes. Although we focused on $T = 1$ capsids due to their simplicity and the uniformity of interfaces across all subunits, the framework for describing subunit shapes proposed is not limited to $T = 1$ capsids. It can be naturally extended to capsids with other triangulation that follows CK theory, that is, when $T = h^2 + hk + k^2$ for integers $(h, k)$ (see Supplementary Fig. 1 and text).

We believe that an explicit description approach of tile shapes and connectivity leads to a new theory of capsid structure that falls outside of CK theory. For example, in the $T = 2$ case, even though the shape of each tile is nearly identical, the connectivity between the tiles is not unique. Non-CK tiling is described as non-isogonal monogonal tiling on a sphere. For the L-A virus (PDB ID 1M1C), we used the dimer as the subunit to be assembled. Since each subunit consists of two identical tiles, the whole system comprises a $T = 2$ assembly. This suggests that analysis and design of nested tiles, that is,

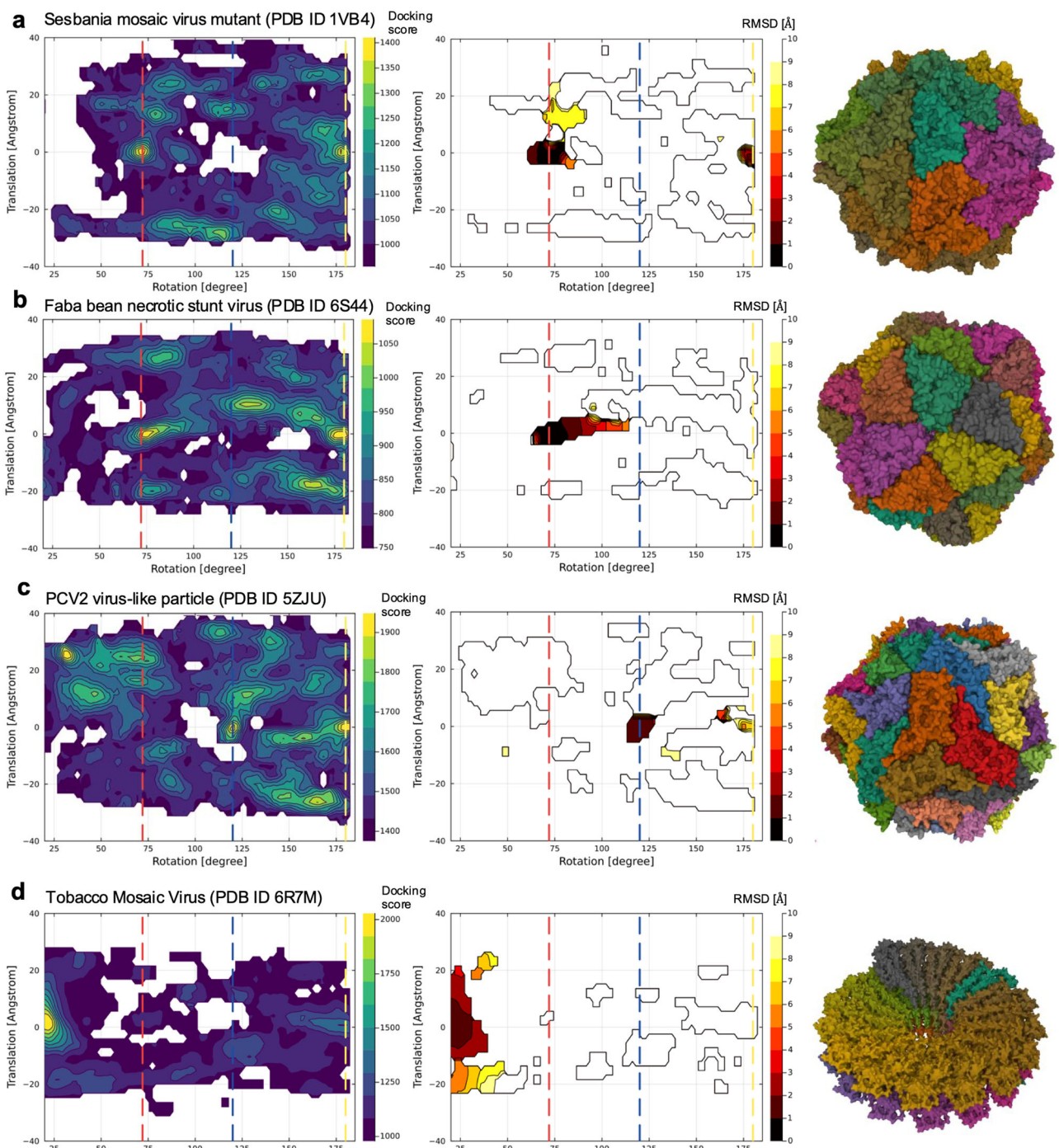

**Fig. 4 | Results of rigid body docking simulations for a pair of subunits. a** Sesbania mosaic virus mutant (PDB ID 1VB4), **b** Faba bean necrotic stunt virus (PDB ID 6S44), **c** Porcine circovirus 2 like particle (PDB ID 5ZJU), and **d** Tobacco mosaic virus (PDB ID 6R7M). Left: filled contour plot of the maximum docking scores of docked poses in the space of rotation and translation of detected screw motions. Vertical dashed lines indicate the rotation positions of 72° (red), 120° (blue), and 180° (yellow), which correspond to the 5-fold, 3-fold, and 2-fold axes, respectively. Middle: filled contour plot of the minimum root mean square displacements of docked poses from the experimental capsid structure. Right: real experimental capsid and non-capsid structures. Different subunits are represented by different colors. For visual clarity, the contours were smoothed by averaging over ~5°.

the division of a single tile further into multiple identical units, may lead to a new way of understanding non-CK cases, such as $T = 2,5,6,\ldots$.

The explicit description of subunit shape developed in this study also assists in the rational design of icosahedral protein complexes. In the design of icosahedral complexes, it is typical to first optimize the interactions either around the 3-fold or the 5-fold symmetry, and then stabilize the interactions around the remaining symmetry axes and the 2-fold axis. Generally, the choice of which symmetry axis (3-fold or 5-fold) to prioritize is not obvious. However, employing the method of our study allows for the determination of whether the 3- or 5-fold axis is more likely to stabilize from a given monomer structure. Moreover, our approach could potentially be extended to describe more complex icosahedral structures, such as those with multiple openings. This extension might involve considering strategic cut-outs in the dihedron, allowing for the modeling of cage-like structures with varied

porosity. Such an adaptation could broaden the applicability of our method to a wider range of nanocontainers. Furthermore, our approach enables the proposal of specific subunit shapes that satisfy spherical tiling for each symmetry axis. These tiling shapes can then be targeted for backbone design and sequence optimization using diffusion model-based frameworks, such as RFDiffusion[34] and Chroma[35], potentially enhancing the design process of icosahedral complexes.

Finally, we discuss the limitations of the proposed framework. The inherent constraint of this framework, based on spherical tiling, is limited to only 2D subunit shapes. There are cases in capsid structures where the shell thickness formed by the subunits is substantial relative to its radius, and the contributions from the three-dimensional interactions at the interface cannot be ignored, e.g., the model of haliangium ochraceum encapsulin (PDB ID 7ODW) in this study. In such cases, the stability of complexes likely affected by not only by the two-dimensional shape but also by three-dimensional shape complementarity between neighboring subunits. For these cases, an extension of the framework would be necessary, such as considering spherical tilings on spheres of various radii and addressing the cumulative effects of these tilings.

## Methods
### Data set of viral capsid structures
The capsid structures of $T = 1$ number were manually curated and selected according to the seqeuence variations and the experimental resolutions from VIPERdb version 3[36]. We used atomic coordinates provided by VIPERdb because they are aligned along the symmetry axis and the alignment is necessary for our analysis. The selected structures include the satellite tobacco necrosis virus (PDB ID 2BUK[19] and 4V4M[20]), the porcine circovirus 2 (3R0R[23], 5ZJU[24]), the faba bean necrotic stunt virus (6S44[21]), the Haliangium ochraceum encapsulin (7ODW[22]), the satellite Panicum Mosaic Virus (1STM[25]), Sesbania mosaic virus deletion mutant (1VB4[26]). Also, the structure $T = 1$ number comprising 120-homomers was taken from the L-A virus (PDB ID 1M1C[27]). For use as a control reference, two non-capsid structures were also used. One is the tobacco mosaic virus (PDB ID 6R7M[37] that has a lockwasher shaped ring with 16 1/3 subunits per turn. The other is the chymotrypsin inhibitor (PDB ID 2M99[38]) which is supposed to exist as a monomer in the physiological condition, lacking any symmetries in interacting modes with other monomers.

The subsequent analysis relies on the atomic coordinates of capsid structures obtained from the VIPERdb database. These coordinates have been pre-aligned to conform to a standardized icosahedral convention, known as the VIPER convention[39], which ensures consistency in the orientation of the capsid structures. In this convention, two icosahedral 2-fold axes coincide with the $z$ and $x$ coordinate axes, while 3-fold and 5-fold axes lie between the $z$ and $x$ axes in the $xz$ plane.

### Fitting of dihedron-unfolding model to experimental capsid structures
We here describe an optimization method to find a junction point that approximates the shape of the capsid from its point data, the radii of amino acids, and the symmetry axes. First, to compare the (normalized) 3D coordinates of the Cα atoms and tile shapes (unfolded dihedron) on a sphere, we use their gnomonic projections (from the center to $z = 1$ plane) $\boldsymbol{p}_i$ and $R_{di}$, respectively. First, we judged the chirality (left- or right-handed) by computing the maximum of minimum distances from the points (the projected positions of the Cα atoms) to the projected axes $X_j^\sigma (j = 2, 3, 5)$, where $\sigma = +1, -1$ represents the left- or right-handed chirality.

$$\text{argmin}_\sigma \max_j \min_i \text{dist}\left(\boldsymbol{p}_i, X_j^\sigma\right)$$

We created a region $R_{ca}$ as the union of a circle of the average radius $r$ of amino acid centering $\boldsymbol{p}_i$. Then, we maximized the Dice coefficient[18] between $R_{ca}$ and the tile shape $R_{di}(x, y)$ computed from the junction point coordinates $(x, y)$. The Dice coefficient between the two regions is defined as:

$$D\left(R_{ca}, R_{di}(x, y)\right) = \frac{2\left|R_{ca} \cap R_{di}(x, y)\right|}{\left|R_{ca}\right| + \left|R_{di}(x, y)\right|}$$

This coefficient ranges from 0 to 1, with 1 indicating perfect overlap. We chose this metric for its effectiveness in comparing spatial overlap of two regions, the real capsid shape $R_{ca}$ and the tile shape $R_{di}(x, y)$. To maximize the overlap, we minimized the following metric over the junction point coordinates $(x, y)$,

$$\text{argmin}_{x,y}\left(1 - D\left(R_{ca}, R_{di}(x, y)\right) + P(x, y)\right)$$

where $P(x, y)$ is a penalty function to avoid self-intersection and to keep the spherical polygon inside a hemisphere. For the optimization, we used Genetic Algorithm solver *Galapagos*[40] on the 3D CAD Rhinoceros/Grasshopper.

### Pairwise docking simulations and analysis of screw angles
Pairwise rigid-body docking simulations were performed for a pair of subunits taken from the whole capsid structure with ZDOCK[30,31]. Two subunits have identical structure with each other, thus structures obtained with the simulations are homo-dimers. At first, we applied ZDOCK with default settings, which outputs 2000 top docking structures at a rotational sampling of 6° interval, corresponding to 54,000 rotations. However, this setting did not yield sufficient statistics. Thus, we effectively performed dense rotational sampling by conducting 1000 independent docking simulation runs with different initial seeds (used for the randomization of the orientations of initial structures). Finally, by ranking the results of independent docking runs based on docking scores, we obtained top 100,000 docking structures, which were then used for the subsequent analysis.

A spatial displacement of a rigid-body can be represented by a rotation about an axis and a translation along the same axis, which is called a screw motion. Here, to investigate the symmetry of the docking structures, we analyzed the structures in terms of screw motions. The screw axis and translation and rotation around that axis were determined by using the Rodriguez equations. Let $\boldsymbol{p}_1$, $\boldsymbol{q}_1$, and $\boldsymbol{r}_1$ be position vectors of three atoms of monomer 1, and $\boldsymbol{p}_2$, $\boldsymbol{q}_2$, and $\boldsymbol{r}_2$ be position vectors of three atoms of monomer 2. Then, the Rodriguez equations (http://robotics.caltech.edu/wiki/images/f/f3/Rodriguez.pdf) are written by,

$$\boldsymbol{p}_2 - \boldsymbol{p}_1 = \tan\left(\frac{\phi}{2}\right)\boldsymbol{\omega} \times \left(\boldsymbol{p}_2 + \boldsymbol{p}_1 - 2\boldsymbol{\rho}\right) + d^\parallel \boldsymbol{\omega}$$

$$\boldsymbol{q}_2 - \boldsymbol{q}_1 = \tan\left(\frac{\phi}{2}\right)\boldsymbol{\omega} \times \left(\boldsymbol{q}_2 + \boldsymbol{q}_1 - 2\boldsymbol{\rho}\right) + d^\parallel \boldsymbol{\omega}$$

$$\boldsymbol{r}_2 - \boldsymbol{r}_1 = \tan\left(\frac{\phi}{2}\right)\boldsymbol{\omega} \times \left(\boldsymbol{r}_2 + \boldsymbol{r}_1 - 2\boldsymbol{\rho}\right) + d^\parallel \boldsymbol{\omega}$$

Here, $\boldsymbol{\omega}$ is a unit vector parallel to the screw axis, $\boldsymbol{\rho}$ is a vector to a point on the screw axis, $\phi$ is the angle of rotation about the screw axis, and $d^\parallel$ is the translation along the screw axis. We calculated these screw motion parameters for the top 100,000 docking structures.

The RMSDs of the docking structure were evaluated by using the experimental structure as a reference. In the case of $T = 1$ structures, the number of picking a pair structure from a capid is $_{60}P_2 = 60 \times 59 = 3,540$ permutations. We calculated the RMSDs of the top 10,000 docking structures relative to 3540 reference structures, and obtained the minimum RMSD value at each region in the space of translation and rotation of the screw axis.

## Statistics and reproducibility

The genetic algorithm optimization for fitting the dihedron-unfolding model was performed using Galapagos solver on Rhinoceros/Grasshopper. For each viral capsid structure, multiple independent optimization runs were conducted to ensure convergence to consistent solutions, as evidenced by the reproducible junction point locations for structures with (almost) identical sequences (PDB IDs 2BUK/4V4M and 3R0R/5ZJU).

For the pairwise docking simulations, extensive sampling was conducted through 1000 independent ZDOCK runs with different random seeds, generating 100,000 top-scoring docking structures. Each run sampled 54,000 rotations at 6° intervals. To validate the reliability of the docking score distributions, we calculated RMSDs between the top 10,000 docking structures and experimental reference structures. For $T = 1$ capsids, all possible pairwise combinations (3540 permutations) from the 60 subunits were used as reference structures. The consistency between high-scoring regions and low RMSD regions confirmed the robustness of our docking analysis results.

## Reporting summary

Further information on research design is available in the Nature Portfolio Reporting Summary linked to this article.

## Data availability

The atomic structure data used in this study are publicly available in the PDB (https://www.rcsb.org) under IDs: 2BUK, 4V4M, 6S44, 7ODW, 3R0R, 5ZJU, 1STM, 1VB4, 1M1C, 6R7M, and 2M99. Pre-aligned atomic coordinates were obtained from VIPERdb version 3 (https://viperdb.org). All data supporting the findings of this study are available in Zenodo[41].

## Code availability

All custom code used in this study is publicly available in Zenodo[41]. The analyses were performed using ZDOCK version 3.0.2 (https://zdock.wenglab.org/software/) for docking simulations, Rhinoceros/Grasshopper with Galapagos solver for genetic algorithm optimization, and Jupyter notebooks for data processing and visualization.

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

## Acknowledgements

The authors thank Sakura Homma for helpful comments on the manuscript. This work was supported by JSPS KAKENHI (Grant numbers: 20K21380, 22H04954 and 23K18105), and partly supported by MEXT as "Program for Promoting Researches on the Supercomputer Fugaku" (Development and application of large-scale simulation-based inferences for biomolecules JPMXP1020230119).

## Author contributions

T. Tachi conceived the idea of unfolding dihedrons for describing subunit shapes. R.T., S.N., T. Tendo, T.H., T. Tachi and Y.M. performed the research. R.T., S.N., T. Tachi and Y. M. wrote the manuscript. T. Tendo and T. H. edited the manuscript. All authors read and approved the final manuscript.

## Competing interests

The authors declare no competing interests.
