## [Transparent Peer Review file · Communications Biology]

Explicit description of viral capsid subunit shapes by unfolding dihedrons

Corresponding Author: Dr Yasuhiro Matsunaga

Version 0:

Reviewer comments:

Reviewer #1

(Remarks to the Author)

The authors present a novel framework for describing the shapes and interfaces of protein subunits in icosahedral viral capsids using spherical tiling theory. This approach addresses the limitations of Caspar-Klug (CK) theory, which does not fully account for subunit shapes or interfaces, both of which are essential for understanding self-assembly. The proposed framework, based on the unfolding of dihedrons with triangular units, systematically characterizes all possible 2D shapes and interfaces of subunit proteins, with a specific focus on $T=1$ capsids. Their findings highlight distinct interface types around symmetry axes, suggesting that these variations play a critical role in regulating subunit interactions. The authors further support their approach through molecular docking simulations, demonstrating a strong correlation between predicted and computed interfaces. This suggests the presence of multiple interaction types during capsid assembly and offers a potential strategy for nucleation in the self-assembly process. Overall, this study makes a novel theoretical contribution, expanding beyond CK theory and providing new insights into the assembly of icosahedral capsids and the rational design of nanocontainers. However, I have several questions that I would like the authors to address before recommending the paper for publication.

1. While I understand the word limits, the brevity in each section makes it difficult to fully follow the paper. In particular, the authors should consider expanding the figure captions, as they are currently too brief, making it hard to fully understand the figures.
2. The authors should provide an explanation of "pairwise docking simulations" in the introduction for better clarity.
3. The authors should define the Dice coefficient, as not all readers may be familiar with this term. Clear definitions will enhance the accessibility of the paper.
4. The following sentence is vague and needs to be clarified: "Consider that the tile is made of a thin sheet; then stitching the identical boundary curves together yields a dihedral shape, precisely, a double-covering of the spherical triangles formed between the three axes of symmetries." The authors should explain this concept more clearly for better understanding.
5. I am unclear on the authors' argument regarding nucleation theory. They state, "Generally, the 5-fold axis is considered a candidate for the critical nucleus due to the greater stability resulting from increased inter-subunit contacts." However, there is often a hexamer at the three-fold axis, not a trimer. While this theory works for $T=1$ capsids, it may not apply to other T numbers. Additionally, pentamers are not generally the most stable capsomers, so why do the authors highlight them? There are always only 12 pentamers in a viral shell, but far more hexamers. Furthermore, other forces and interactions, beyond hydrophobic interactions, influence assembly—such as how proteins are stretched or bent in specific positions, See for example ACS Nano 14, 3170-3180 (2020). The authors should clarify these points and explain the source of the energy barrier in their model.

Reviewer #2

(Remarks to the Author)

Please see attached pdf review.

Reviewer #3

(Remarks to the Author)

Toyooka et al. present a novel approach to describe 2D shapes and interfaces on icosahedral capsid structures. The

introduction sounds solid and comprehensive, and the current state of theorems is explained. The authors explain their dihedral-unfolding model in detail, and this is underpinned by two descriptive figures and three supplementary movies. Due to my profession, I cannot comment on the reliability and impact of their mathematical description of their model. The authors applied their model to various real $T = 1$ viral capsid structures from the PDB. The selection includes various viral structures with two duplicates, which is rationalized in the text. The logic of the selection of the viral structures is not completely clear. E.g., the structure of the Sesbanian mosaic virus capsid (1VB4) is neither the highest resolution nor the oldest structure. Additionally, it contains a mutation. I am curious why the authors do not use the native viral structure, which also has better resolution (PDB 1SMV). Also, for the $T = 2$ L-A helper virus, there is a newer structure with a higher resolution (PDB 8PE4). The monomeric subunit docking using ZDOCK is very robust, and the results are presented well. It is very nice to see that these results confirm the mathematical approach of the authors. Also, the exceptional cases 7ODW and 1M1C are well described and leave space for a follow-up manuscript dealing with more complex structures ($T \neq 1$), which they also highlight in their discussion.

Major comments:

- The manuscript shows an inconsistency in the figure panel description and how they are addressed in the text. While Figures 1 & 2 panels are referenced with left, middle, right, Figures 3 & 4 panels are addressed with a, b, c, d. The authors should harmonize this, and I suggest using lower case letters as they do in Figures 3 & 4.
- The figure legends are very short, and it's really hard to understand the figures. Also, not every color/line is well described. The figure legend itself should be sufficient to understand the figure, and unfortunately this is not the case in this manuscript.
- Figure 4. The x- and y-axes of the two heatmaps should be equal to have a better visual comparison. This is especially very obvious in panel d. The y-axis on the docking scoring looks like -29 to 29, whereas the RMSD heatmap is more like -25 to 25. On the x-axis, the effect is even more pronounced. Please have the exact same y- and x-axis boundaries. Additionally, the panels from a to d should be properly aligned. In a, the RMSD plot is shifted to the left.

Minor comments:

- p. 5, lines 140-141 & Figure 3: See Fig. 3a for the interpretation of the criteria. Unfortunately, the figure does not really explain the criteria. For me, it's not really clear how and which shortest distance was chosen. Also, the figure panel could have more description; e.g., I can only assume the arrows labelled 2, 3 & 5 are the symmetry axes. Also, the blue line is not described. Additionally, panel B should have small headlines saying 'left-handed' and 'right-handed', as this is only described in the text (see also general Major comment on figure legends).
- p. 6, line 153. The sentence "the structure of 1M1C" should start with an uppercase T.
- Inconsistent usage of protein names and PDB IDs. I suggest to use the protein/capsid names, rather than using the PDB IDs as names. As I assume, the authors are not virologist/biologist, and thereby the usage of unique identifiers might be easier, but biologist rarely use PDB IDs as in-text identifiers. So instead of writing "The structure of 1M1C25, classified as $T = 2$ comprising a 120-homomer, was also included." (p. 6, line 153), the authors should write "The structure of the L-A helper virus (PDB ID 1M1C), classified as [..]".

Out of curiosity, can this new method also be applied to other icosahedral structures like the pyruvate dehydrogenase complex (i.e., PDB 8PIU), as not only closed capsids might be interesting as nanocontainers, but also this icosahedral cage with a lot of openings.

To summarize, the manuscript presented is, after some revisions, suitable for publication in Communications Biology.

Version 1:

Reviewer comments:

Reviewer #1

(Remarks to the Author)

The authors have modified the paper based on my recommendation and addressed all my concerns. Thus I recommend the paper for publication.

Reviewer #2

(Remarks to the Author)

The authors have addressed all the points I raised, and have enhanced the quality of the manuscript by clarifying the discussion of spherical tiling theory and by adding an additional informative figure. I believe that the manuscript is now ready for publication in Communications Biology.

Reviewer #3

(Remarks to the Author)

The authors have addressed all my concerns. All figures and figure legends are now of high quality and will help the increase the impact of this manuscript. Additionally, the rationale of choosing the atomic structures is now clear - I was not aware, that there is an additional alignment procedure in the ViperDB, which is now clearly stated.

Also, I like to appreciate, that the authors included the outlook for partially open icosahedral structures and I am looking forward towards a follow up manuscript.

Responses to the comments

Reviewer 1

We first sincerely thank the reviewer for his/her comments and suggestions.

> 1. While I understand the word limits, the brevity in each section makes it difficult to fully follow the paper. In particular, the authors should consider expanding the figure captions, as they are currently too brief, making it hard to fully understand the figures.

We appreciate your feedback and agree that the figure captions were too brief. We have significantly expanded all figure captions to provide more detailed explanations. For example, the caption for Figure 3 now reads:

“Figure 3. Analysis of real capsid structures using our dihedral unfolding framework. a. Fitting of the junction point location to a real capsid structure. Gnomonic projection of a subunit is shown for the L-A helper virus, PDB ID 1M1C. Black dots represent the C α atom positions of the subunit protein. Arrows labeled 2-fold, 3-fold, and 5-fold indicate the projected 2-fold, 3-fold, and 5-fold symmetry axes, respectively. The blue line outlines the spherical and projected triangles (dihedrons). Red dots represent the locations of the junction points on the projected plane that best matches our model to the real structure. The orientation (chirality) of tile is determined according to the distances between the symmetry axes and the C α atom positions. The junction point location is optimized by maximizing the dice coefficient. b. Results of fitting our model to various real capsid structures. Left and right panels show left-handed and right-handed orientations. Colored regions indicate where the junction point does not induce self-intersection, with yellow, blue, and red areas corresponding to domains governed by contact with respect to the 2-, 3-, and 5-fold axes, respectively. Capsid structures (black circles) are labeled with their PDB IDs. c. Examples of subunit shapes and fitted tile shapes for the faba bean necrotic stunt virus (PDB ID 6S44), and PCV2 virus-like particle (PDB ID 5ZJU).”

Similar expansions have been made to all other figure captions to ensure they are self-contained and informative.

> 2. The authors should provide an explanation of "pairwise docking simulations" in the introduction for better clarity.

Thank you for this suggestion. We have added the following explanation to the introduction:

“To validate our classification and understand the physical interactions between subunits, we employ pairwise docking simulations. These simulations investigate the potential interactions between two protein subunits, predicting their most favorable binding orientations and energies. By comparing the results of these simulations with our geometric classification, we can assess the consistency between the predicted physical interactions and the structural patterns identified by our framework.”

> 3. The authors should define the Dice coefficient, as not all readers may be familiar with this term. Clear definitions will enhance the accessibility of the paper.

We appreciate this point and have added a definition of the Dice coefficient in the Methods section:

“Then, we maximized the Dice coefficient¹⁸ between R_{ca} and the tile shape $R_{di}(x, y)$ computed from the junction point coordinates (x, y) . The Dice coefficient between regions X and Y is defined as:

$$D(X, Y) = \frac{2|R_{ca} \cap R_{di}(x, y)|}{|R_{ca}| + |R_{di}(x, y)|}$$

This coefficient ranges from 0 to 1, with 1 indicating perfect overlap. We chose this metric for its effectiveness in comparing spatial overlap of two regions, the real capsid shape R_{ca} and the tile shape $R_{di}(x, y)$.”

In the text above, we have added a citation (Ref. 18) to a paper about the Dice coefficient (Taha, A. A. & Hanbury, A. Metrics for evaluating 3D medical image segmentation: analysis, selection, and tool. *BMC Med Imaging* **15**, 29 (2015)).

> 4. The following sentence is vague and needs to be clarified: "Consider that the tile is made of a thin sheet; then stitching the identical boundary curves together yields a dihedral shape, precisely, a double-covering of the spherical triangles formed between the three axes of symmetries." The authors should explain this concept more clearly for better understanding.

We appreciate the reviewer's comment regarding the clarity of our explanation. We have thoroughly revised this section to provide a more precise and detailed description of the stitching process and its result. The updated text now reads:

“Consider the tile as a thin, flexible sheet on the sphere's surface. The symmetry axes of the icosahedron ensure that when we stitch the identical boundary curves together, they align perfectly (Figs. 1b and 1c). As we stitch along each symmetry axis, the vertices of the tile are brought closer together. The final stitching necessarily brings all three vertices to meet at a single intersection point. In the end, this stitching transforms the original tile into a *dihedron*¹⁶: a doubly-covered half equilateral triangle that covers two adjacent triangular regions. This dihedron is a 2-to-1 mapping precisely covering 1/120 of the sphere's surface.”

This revised explanation clarifies the following key points:

1. The role of icosahedral symmetry in ensuring perfect alignment during stitching.
2. The step-by-step process of how the vertices converge to a single point.
3. The transformation of the original tile into a doubly-covered half equilateral triangle (dihedron).

Additionally, we have added a citation to a mathematical book on tiling theory (Ref. 16) (Akiyama, J. & Matsunaga, K. *Treks into Intuitive Geometry: The World of Polygons and Polyhedra*. (Springer, 2015)) that covers the subject of unfolding dihedrons.

> 5. I am unclear on the authors' argument regarding nucleation theory. They state, "Generally, the 5-fold axis is considered a candidate for the critical nucleus due to the greater stability resulting from increased inter-subunit contacts." However, there is often a hexamer at the three-fold axis, not a trimer. While this theory works for T=1 capsids, it may not apply to other T numbers. Additionally, pentamers are not generally the most stable capsomers, so why do the authors highlight them? There are always only 12 pentamers in a viral shell, but far more hexamers. Furthermore, other forces and interactions, beyond hydrophobic interactions, influence assembly—such as how proteins are stretched or bent in specific positions, See for example ACS Nano 14, 3170-3180 (2020). The authors should clarify these points and explain the source of the energy barrier in their model.

We appreciate the reviewer's insightful comments regarding nucleation theory and the factors influencing capsid assembly. We acknowledge that our original statement was oversimplified, and we have revised our discussion to more accurately reflect the complexities of capsid assembly, including the formation of both pentamers and hexamers, as studied ACS Nano 14, 3170-3180 (2020). We agree that multiple factors contribute to capsid assembly, including elastic energy as revealed in the ACS Nano paper. We have revised our text to incorporate these concepts and provide a more comprehensive explanation of the assembly process.

The updated text now reads:

“The initial stages of capsid self-assembly involve complex interactions between subunits. Our pairwise docking simulations indicate that there may be preferences in subunit interactions near different symmetry axes depending on the subunit shape. Notably, our results suggest that dimers around the 5-fold or 3-fold axes can be stabilized without the presence of other neighboring contacts. This finding provides insights into the potential early stages of capsid assembly, although it does not necessarily imply that dimer formation is the most important step in the process.

It is noted that capsid assembly is influenced by multiple forces beyond simple hydrophobic interactions. As recently studied by Panahandeh et al., elastic energy from protein stretching and bending significantly affects the assembly process and final capsid structure. These elastic forces, along with hydrophobic interactions and other factors such as protein concentration, contribute to the energy barriers in capsid formation and help guide the assembly towards symmetrical structures. For capsids with T numbers greater than 1, the assembly process generally involves the formation of both pentamers and hexamers. While pentamers are necessary due to the spherical geometry of the capsid, hexamers are generally preferred by hydrophobic interactions. The interplay between these structures is crucial for the final capsid formation.”

Responses to the comments

Reviewer 2

We first sincerely thank the reviewer for his/her comments and suggestions.

> Main points:

> The “Dihedron-unfolding model based on spherical tiling” subsection in the “Results” section is rather technical. In particular there is a lot of geometric “jargon” that will likely be unfamiliar to a general biology audience (and indeed to most readers, who are not necessarily well-versed in the details of spherical tiling theory.) Phrases like “tree graph,” “homeomorphic,” and “topological disk” will not be familiar to a general audience and should be explained (or alternatively, some of these technical terms could be removed or placed in descriptions in the SI.)

We appreciate your feedback regarding our technical description of the "Dihedron-unfolding model based on spherical tiling" subsection. We understand your concern about the geometric terminology potentially being challenging for a general biology audience. In response to your comments, we have carefully revised this section to improve its accessibility while maintaining the essential technical information.

Our revision includes the following changes:

1. We have removed or replaced some of the more specialized geometric terms. For example, “tree graph” has been replaced with "Y-shaped path," and "topological disk" has been removed entirely. The term "homeomorphic" has been replaced with "classified to," which better describes the relationship between the tilings and the polyhedra without requiring knowledge of topological concepts.
2. We have provided brief, clear explanations for necessary technical terms. For instance, "geodesics" are now explained parenthetically as "shortest paths on the sphere's surface."
3. We have improved the clarity of our explanations by relating geometric concepts directly to the figures. For example, we now explicitly refer to the polyhedral graph and its dual graph as they appear in Figure 2, making these concepts more concrete for readers.
4. We have maintained the overall structure of the explanation but have used clearer language to guide readers through the concepts. Particular attention has been paid to explaining the significance of the junction point and how it relates to tile shapes and connections.
5. For each type of tiling, we now provide a brief description of the resulting polyhedron and its dual, using more comprehensive language (the number of faces and their shapes). This approach helps readers visualize these complex structures without requiring extensive knowledge of polyhedron theory.

> The observation in Fig. 3 that proteins fall into different classes of 2-fold, 3-fold, and 5-fold axis dominant interactions is interesting, but it is a little difficult to understand without more visuals. Would it be possible to show visual examples or schematics of subunit-subunit interactions for these different classes, perhaps in another panel below 3b?

Thank you for this suggestion. We agree that additional visuals would enhance the understanding of these different interaction classes. In this revision, we have added a new panel (Fig. 3c) that provides schematic representations of subunit-subunit interactions for 3-fold, and 5-fold axis dominant interactions. This new visual aid should help readers better grasp the geometric differences between these interaction types.

> Minor points:

> I am confused what is meant by “polyhedron with three vertices” on l. 79. Isn’t the minimum number of vertices in a polyhedron 4?

Thank you for pointing this out. The reviewer is correct that a standard polyhedron has a minimum of 4 vertices. Our original wording referred to a dihedron, which is a special case with 3 vertices. We think this terminology may be confusing for readers of Communications Biology. So, we have revised the text to focus on the unfolding process without delving into the specifics of the dihedron's structure.

The revised sentence now reads:

"To unfold the triangular dihedron into a tile, the dihedron must be cut along a Y-shaped path (cut tree) starting from a single point we call the *junction point* with three ends at the vertices (the cut tree and the junction point shown in Figs. 1b and 1c)"

> Please add a citation for the Dice coefficient on l. 141, I suspect few readers will be familiar with this metric.

Thank you for pointing this out. We have added a citation for the Dice coefficient (Taha, A. A. & Hanbury, A. Metrics for evaluating 3D medical image segmentation: analysis, selection, and tool. BMC Med Imaging 15, 29 (2015)) and provided additional explanation of the metric in the Methods section. We have added a schematic figure in Fig. 4 which explains the idea of the Dice coefficient.

> In Fig. 3a, please indicate what protein is being shown.

We have updated the caption of Fig. 3a to explain that the example shown is from the L-A helper virus (PDB ID 1M1C).

> I noticed several typos when reading through the manuscript – please give it a quick proofread.

We apologize for the typos in the previous version of the manuscript. We have carefully proofread the entire manuscript and corrected identified errors.

Responses to the comments

Reviewer 3

We first sincerely thank the reviewer for his/her comments and suggestions.

> The authors applied their model to various real $T = 1$ viral capsid structures from the PDB. The selection includes various viral structures with two duplicates, which is rationalized in the text. The logic of the selection of the viral structures is not completely clear. E.g., the structure of the Sesbanian mosaic virus capsid (1VB4) is neither the highest resolution nor the oldest structure. Additionally, it contains a mutation. I am curious why the

authors do not use the native viral structure, which also has better resolution (PDB 1SMV). Also, for the T = 2 L-A helper virus, there is a newer structure with a higher resolution (PDB 8PE4).

We appreciate the reviewer's suggestions. Regarding the Sesbania Mosaic Virus (1SMV), it is indeed a nice structure. However, as it belongs to T=3 group, it falls outside the scope of our current study focusing on T=1 structures. We are going to study it as part of our future work when we extend our approach beyond T=1 structures. Concerning the LA virus (8PE4), you are correct that it offers a higher resolution. However, due to its recent publication, it is not yet registered in the VIPERdb database that we utilized for our study. Consequently, we do not have access to the capsid structure coordinates aligned along the symmetry axes (VIPER coordinates) for this entry. While it is possible for us to align the structure ourselves, this could potentially compromise the consistency and reproducibility of our calculations across different structures.

In response to the reviewer's comments, we have added an explanation in the Methods section clarifying our criteria for capsid structure selection, including the requirement for available VIPER coordinates. The added text reads as follows:

“The capsid structures of T=1 number were manually curated and selected according to the sequence variations and the experimental resolutions from VIPERdb version 3. We used atomic coordinates provided by VIPERdb because they are aligned along the symmetry axis and the alignment is necessary for our analysis.”

> Major comments:

> • The manuscript shows an inconsistency in the figure panel description and how they are addressed in the text. While Figures 1 & 2 panels are referenced with left, middle, right, Figures 3 & 4 panels are addressed with a, b, c, d. The authors should harmonize this, and I suggest using lower case letters as they do in Figures 3 & 4.

We thank the reviewer for pointing out this inconsistency. We have revised all figure panel descriptions to consistently use lowercase letters (a, b, c, d) throughout the manuscript.

> • The figure legends are very short, and it's really hard to understand the figures. Also, not every color/line is well described. The figure legend itself should be sufficient to understand the figure, and unfortunately this is not the case in this manuscript.

We agree that our figure legends were insufficiently detailed. We have substantially expanded all figure legends to make them self-contained, explaining all colors, lines, and symbols used. Additionally, we have substantially expanded all figure captions as described in our response to Reviewer 1's first comment.

> • Figure 4. The x- and y-axes of the two heatmaps should be equal to have a better visual comparison. This is especially very obvious in panel d. The y-axis on the docking scoring looks like -29 to 29, whereas the RMSD heatmap is more like -25 to 25. On the x-axis, the effect is even more pronounced. Please have the exact same y- and x-axis boundaries. Additionally, the panels from a to d should be properly aligned. In a, the RMSD plot is shifted to the left.

We appreciate this detailed feedback on Figure 4. We have revised all the plots to ensure that the x- and y-axes of all heatmaps have the same scale and range for better visual comparison. We have also realigned all panels to ensure they are properly positioned relative to each other. The same type of plots in all supplementary figures (Figs. S6 and S7) have been revised as well.

> Minor comments:

> • p. 5, lines 140-141 & Figure 3: See Fig. 3a for the interpretation of the criteria. Unfortunately, the figure does not really explain the criteria. For me, it's not really clear how and which shortest distance was chosen. Also, the figure panel could have more description; e.g., I can only assume the arrows labelled 2, 3 & 5 are the symmetry axes. Also, the blue line is not described. Additionally, panel B should have small headlines saying 'left-handed' and 'right-handed', as this is only described in the text (see also general Major comment on figure legends).

>

> • p. 6, line 153. The sentence "the structure of 1M1C" should start with an uppercase T.

>

> • Inconsistent usage of protein names and PDB IDs. I suggest to use the protein/capsid names, rather than using the PDB IDs as names. As I assume, the authors are not virologist/biologist, and thereby the usage of unique identifiers might be easier, but biologist rarely use PDB IDs as in-text identifiers. So instead of writing "The structure of 1M1C25, classified as $T = 2$ comprising a 120-homomer, was also included." (p. 6, line 153), the authors should write "The structure of the L-A helper virus (PDB ID 1M1C), classified as [..]".

We thank the reviewer for these detailed observations. We have addressed each of these minor comments:

- We have improved the explanation of the criteria in Figure 3a and its caption.
- We have corrected the capitalization in "The structure of 1M1C" (p. 6, line 153).

- We now consistently use protein/capsid names instead of PDB IDs when referring to structures in the text.

> Out of curiosity, can this new method also be applied to other icosahedral structures like the pyruvate dehydrogenase complex (i.e., PDB 8PIU), as not only closed capsids might be interesting as nanocontainers, but also this icosahedral cage with a lot of openings.

Thank you for this important suggestion. The application of our method to icosahedral structures with numerous openings, such as the pyruvate dehydrogenase complex (PDB ID 8PIU), is indeed an exciting prospect. It might be possible that we could describe such structures by considering cut-outs in the dihedron. However, the efficacy of this approach needs to be thoroughly investigated. In our future research, we are interested in exploring this possibility both theoretically and practically. We aim to examine whether our method can be adapted to accurately model and fit existing cage structures with multiple openings.

Inspired by the reviewer's suggestion, we have added the following sentences to the Discussion section of our manuscript:

“Moreover, our approach could potentially be extended to describe more complex icosahedral structures, such as those with multiple openings. This extension might involve considering strategic cut-outs in the dihedron, allowing for the modeling of cage-like structures with varied porosity. Such an adaptation could broaden the applicability of our method to a wider range of nanocontainers.”

Sincerely yours,

Yasuhiro Matsunaga

Yasuhiro Matsunaga, D.Sci., Associate Prof.,
Saitama University
255 Shimo-Okubo, Sakura-ku, Saitama, 338-8570, Japan
E-mail: ymatsunaga@mail.saitama-u.ac.jp
Phone: +81-48-858-9210